# Cathepsin S Knockdown Suppresses Endothelial Inflammation, Angiogenesis, and Complement Protein Activity under Hyperglycemic Conditions In Vitro by Inhibiting NF-κB Signaling

**DOI:** 10.3390/ijms24065428

**Published:** 2023-03-12

**Authors:** Shithima Sayed, Omar Faruq, Umma Hafsa Preya, Jee Taek Kim

**Affiliations:** 1Department of Ophthalmology, College of Medicine, Chung-Ang University, Seoul 06974, Republic of Korea; shithima.s.s.111@gmail.com (S.S.);; 2Biomedical Engineering Doctoral Program, Boise State University, Boise, ID 83725, USA; 3Department of Ophthalmology, Chung-Ang University Hospital, Seoul 06974, Republic of Korea

**Keywords:** cathepsin S, hyperglycemia, inflammatory cytokines, cathepsin S siRNA, transfection, HUVECs

## Abstract

Hyperglycemia plays a key role in the development of microvascular complications, endothelial dysfunction (ED), and inflammation. It has been demonstrated that cathepsin S (CTSS) is activated in hyperglycemia and is involved in inducing the release of inflammatory cytokines. We hypothesized that blocking CTSS might alleviate the inflammatory responses and reduce the microvascular complications and angiogenesis in hyperglycemic conditions. In this study, we treated human umbilical vein endothelial cells (HUVECs) with high glucose (HG; 30 mM) to induce hyperglycemia and measured the expression of inflammatory cytokines. When treated with glucose, hyperosmolarity could be linked to cathepsin S expression; however, many have mentioned the high expression of CTSS. Thus, we made an effort to concentrate on the immunomodulatory role of the CTSS knockdown in high glucose conditions. We validated that the HG treatment upregulated the expression of inflammatory cytokines and CTSS in HUVEC. Further, siRNA treatment significantly downregulated CTSS expression along with inflammatory marker levels by inhibiting the nuclear factor-kappa B (NF-κB) mediated signaling pathway. In addition, CTSS silencing led to the decreased expression of vascular endothelial markers and downregulated angiogenic activity in HUVECs, which was confirmed by a tube formation experiment. Concurrently, siRNA treatment reduced the activation of complement proteins C3a and C5a in HUVECs under hyperglycemic conditions. These findings show that CTSS silencing significantly reduces hyperglycemia-induced vascular inflammation. Hence, CTSS may be a novel target for preventing diabetes-induced microvascular complications.

## 1. Introduction

Hyperglycemia and hyperglycemia-induced vascular complications are associated with increased risks of high blood pressure, coronary heart disease, and strokes [1,2]. Hyperglycemia has direct impacts on vascular systems, and vascular complications are a major cause of morbidity and mortality in diabetic patients. High levels of glucose elevate the flux of sugar through the polyol pathway, leading to intracellular sorbitol accumulation that increases osmotic stress [3]. The development of diabetic complications has been linked to a number of pathophysiologic pathways, including the polyol pathway, non-enzymatic glycation, oxidative stress, and activation of protein kinase C. These mechanisms are mostly associated with excessive glucose transport into retinal cells [4]. Furthermore, inflammation plays a key role in the development of diabetic complications [5]. Micro- and macrovascular diabetic complications are largely due to long-term hyperglycemia, in addition to other major complications [2]. Retinopathy, nephropathy, and diabetic vascular diseases are due to the effects of high blood sugar on retinal, endothelial, and mesangial cells [6]. Microvascular endothelial cells can potentially be damaged in hyperglycemia since they are unable to downregulate the rate of glucose transport, resulting in elevated concentrations of glucose within the cells [7]. This leads to microvascular complications, including increased leukocyte adhesion, permeability, and procoagualant activity [7,8]. Furthermore, hyperglycemia-induced disruptions of vascular homeostasis in endothelial cells promote the release of inflammatory cytokines as well as the production of free radicals and reactive oxygen species (ROS) [9]. As a result, cellular damage and inflammation often progress in hyperglycemic states. Moreover, vascular remodeling via extracellular matrix (ECM) degradation is explicitly responsible for microvasculature alterations and neovascularization. In the early stages of diabetic retinopathy, microaneurysms have been observed, whereas, in the final stages, angiogenesis and fibrovascular proliferation are commonly seen [10].

Cathepsin S (CTSS) is a lysosomal protease and a unique member of the cysteine cathepsin protease family. It is involved in a variety of pathological processes, including cancer, cardiovascular disease, and arthritis. It has been demonstrated that CTSS expression is significantly higher in diabetes patients, and it serves as a robust biomarker for diabetes and atherosclerosis [11,12]. Further, it has a critical role in vascular remodeling via ECM degradation as an elastolytic protease and is related to severe microvascular complications [13,14]. Moreover, CTSS is associated with plasma leakages, which contribute to vascular wall degeneration and microvasculature damage [15]. Previous studies have reported that CTSS is involved in the inflammatory processes of immune diseases such as atopic dermatitis, psoriasis, bronchial asthma, and rheumatoid arthritis [16,17], as well as in autoantigen presentation in the immune system [18]. In recent research, CTSS has been shown to play an important role in invasion, metastasis, and tumorigenesis [19]. Furthermore, it has been linked to the development of type 2 diabetes, where it is elevated at both the mRNA and protein levels [20]. CTSS causes chronic inflammation during diabetic complications that is responsible for neurovascular degeneration [21,22] and promotes microvascular complications. CTSS activities were associated with the macrophage subpopulation in NOD mice and in human type 1 diabetic samples [23]. Cathepsin silencing reduces hyperglycemia-mediated complications and delays the progression of diabetes [24]. However, few studies have observed the relationship between silencing CTSS and vascular complications under hyperglycemia.

Recent research has focused on understanding and determining the effects of hyperglycemia on specific types of endothelial cells (ECs) or vasculature components. ECs are found in various organs and vessels, with different functions depending on their location [25,26,27]. Hyperglycemic endothelial dysfunction (ED) presents differently in ECs from different parts of the vasculature. ECs in the aorta and lung exhibit differential dipeptidyl peptidase-4 expression, as well as different activity levels under hyperglycemia [28]. In addition, Karabach et al. observed significantly different cell viability and free radical formation in EA.hy.926 and human umbilical vein endothelial cells (HUVECs) under hyperglycemia [29]. Since HUVECs are a promising candidate for the in vitro study of hyperglycemia-induced vascular complications, we investigated the effects of a CTSS knockout in HUVECs in a hyperglycemic environment. The findings of this study may shed light on the importance of CTSS silencing in hyperglycemia and establish effective guidelines for the development of novel therapeutics to treat hyperglycemic-related vascular complications.

## 2. Results

### 2.1. Effects of Different Glucose Concentrations on HUVECs’ Viability

HUVEC viability was evaluated after 24 h of treatment with various concentrations of glucose. Figure 1A shows that after 24 h of treatment with 15 mM and 30 mM glucose, a cell viability of over 80% was observed. In contrast, less than 80% of cells were viable after 24 h of treatment with 60 mM glucose. These results confirm that treatment with 15 mM glucose achieves a cell viability of over 95% and, hence, exerts only a minor hyperglycemic effect on the cells (Figure 1A). Therefore, to induce hyperglycemia in HUVECs, we selected a 30 mM glucose concentration for 24 h for further experiments.

### 2.2. Expressions of CTSS and Other Pro-Inflammatory Cytokines Are Induced by High Glucose (HG) in HUVECs

To study whether CTSS expression was affected by hyperglycemic conditions, HUVECs were treated with the 30 mM glucose treatment for 24 h. The expressions of pro-inflammatory cytokines and CTSS were increased in HG-treated samples compared to those in the controls (Figure 1B), as evaluated by quantitative real-time polymerase chain reaction (qRT-PCR). Western blot (WB) analysis confirmed that CTSS, VEGFA, and other inflammatory markers (Figure 1C) were overexpressed in HG-treated samples than in the controls. Inflammatory markers TNF-α, IL-1β, IL-6, NF-κB, MCP-1, COX-2, VEGFA, and iNOS, and adhesion markers VCAM-1 and ICAM-1 showed higher expression after HG treatment, as revealed by qRT-PCR and WB. These results suggest that treatment with 30 mM glucose successfully mimicked hyperglycemia conditions in HUVECs.

### 2.3. Effects of CTSS Knockdown on HUVECs

The effect of the CTSS knockdown was evaluated using qRT-PCR, WB, and immunostaining in HUVECs post-siRNA treatment. CTSS expression was downregulated after the siRNA treatment (Figure 1D,E). Immunostained images indicate the expression of CTSS in HG-treated HUVECs with and without CTSS siRNA transfection (Figure 1F). CTSS was highly expressed after HG treatment and significantly decreased after CTSS knockdown.

After siRNA transfection of HUVECs with and without HG treatment, the cytotoxic effects of CTSS knockdown were examined (Figure 1G). Over 80% cell viability was observed after 24 h of transfection. Fluorescence micrographs of different HUVEC samples showed similar cell proliferation. These data confirmed that siRNA transfection had no lethal effects on the cells.

### 2.4. CTSS Knockdown Inhibits the Expression of Pro-Inflammatory Markers

To investigate whether silencing CTSS could reduce the expression of pro-inflammatory cytokines in HUVECs, their expressions at the gene (Figure 2A–G) and protein (Figure 2H) level were examined by qRT-PCR and WB. Pro-inflammatory cytokine expression was upregulated in HG-treated samples (siCON + HG) compared to that in the control samples (siCON) and significantly downregulated in the CTSS siRNA-transfected samples (siCTSS). Samples treated with both HG and CTSS siRNA (siCTSS + HG) also showed significant downregulation of the expression of all the studied gene markers. The localization of a major inflammatory cytokine, TNF-α, was visualized by immunostaining and fluorescence microscopy (Figure 2I). TNF-α was expressed at higher levels in the HG-treated sample, while its expression sharply decreased after CTSS siRNA treatment, which is in agreement with the qRT-PCR data (Figure 2I).

### 2.5. CTSS Knockdown Inhibits Vascular Endothelial Growth Factor and Vascular Adhesion Marker Expression

The gene expressions of vascular endothelial growth marker (VEGFA) and vascular adhesion markers (VCAM-1 and ICAM-1) were investigated by qRT-PCR. CTSS knockdown (siCTSS + HG) caused significant downregulation of these markers compared to that in the controls (siCON + HG) (Figure 3A–C). The results of WB protein analysis largely agreed with the gene expression data (Figure 3D). These results (Figure 2 and Figure 3A–D) suggest that HG induces inflammatory cytokine expression while CTSS silencing decreases their expression. CTSS knockdown also significantly reduced the expressions of transcription factors: nuclear factor-kappa B (NF-κB) and inducible NO synthase (iNOS), along with those of VEGFA, ICAM-1, and VCAM-1. In contrast, higher expression was only observed in HG-treated samples (siCON + HG).

### 2.6. CTSS Knockdown Inhibits Angiogenesis

To determine the effects of CTSS siRNA on angiogenesis and vascular remodeling, pretreated HUVECs were seeded on Matrigel to observe tube formation. Hyperglycemia induced tube formation in siCON + HG samples (Figure 3E–I). The CTSS knockdown inhibited tube formation even under hyperglycemic conditions. The inhibitory effects of CTSS siRNA treatment on angiogenesis in SiCTSS and SiCTSS + HG cells can be seen in Figure 3E–I.

### 2.7. CTSS Knockdown Downregulates Complement Activation in HUVECs

To study the associations between CTSS and complement activation in hyperglycemia, the expression patterns of C3, C5, C3a, and C5a were analyzed. C3 and C5 mRNA levels were elevated in HG-treated samples (siCON + HG) but downregulated in samples that underwent siRNA treatment (siCTSS and siCTSS + HG) (Figure 4A,B). C3a and C5a protein levels were also increased in HG-treated samples (siCON + HG) and decreased in samples with CTSS silencing (siCTSS and siCTSS + HG) (Figure 4C). Immunocytochemical staining of C5a (Figure 4D) revealed trends similar to those observed in the WB and qRT-PCR data.

## 3. Discussion

Hyperglycemia is responsible for microvascular damage, inducing a pro-inflammatory state that is the main feature of vasculopathy [30,31]. HUVECs were treated with HG to mimic hyperglycemic conditions in the current study, and hyperglycemia-associated changes in gene expression were assessed. The in vitro HUVEC model is a popular method for studying hyperglycemia-induced vascular complications and identifying new targets. Our study shows that pro-inflammatory cytokines, along with VEGFA and adhesion molecules, were sharply upregulated under hyperglycemic conditions (Figure 1B,C), which is in agreement with previous studies [32,33]. ECs can act as regulators of inflammatory processes and function as innate immune cells under many pathophysiologic conditions [33]. In patients with hyperglycemia, researchers have detected chronic low-grade, subclinical inflammation mediated by TNF-α, COX-2, IL-1β, IL-6, and MCP-1, which is responsible for vascular complications [32,34]. Chronic inflammation contributes to neurodegeneration, pericyte depletion, EC loss leading to ED, and disruptions of the blood-retinal barrier (BRB) [22,32]. Further, the toxic effects of HG are mediated by increased levels of oxidative stress [31] which induce lysosomal leakage through increased lysosomal permeabilization, releasing CTSS into the cytoplasm [35].

Cathepsin expression is influenced by metabolic status and biochemical requirements [15,18,36]. Moreover, CTSS induces protease-activated receptor (PAR)-2 on ECs and leads to ED [37]. Since CTSS is upregulated under high blood glucose levels in vivo and in diabetic rat models [24,38], overexpression of CTSS can be considered a pathological factor in the development of diabetes. We examined CTSS activity in HUVECs under HG conditions before and after CTSS silencing. Our study shows that CTSS siRNA effectively inhibited CTSS expression under HG conditions (Figure 1D–F), leading to reductions in inflammation, angiogenesis, and vascular complications. It has been demonstrated that hyperglycemia induces inflammation that is characterized by the activation of transcription factors, such as NF-κB, as well as increased chemokine and cytokine expression [39]. Higher CTSS levels are linked to inflammatory cytokine release, which leads to the onset of type 2 diabetes. In this investigation, CTSS knockdown significantly reduced HG-induced expression of pro-inflammatory cytokines, including MCP-1, COX-2, iNOS, IL-1β, IL-6, and NF-κB (Figure 2). NF-κB is a pivotal inflammatory mediator that serves as both an innate and an adaptive immune response regulator [40]. In vascular ECs, genes activated by NF-κB include those encoding for various inflammatory chemokines, cytokines, and adhesion molecules [40,41]. Further, this study observed that HG causes the upregulation of inducible enzymes such as iNOS and COX-2, which are also regulated by NF-κB [40,41]. COX-2 and iNOS are known to facilitate HUVEC apoptosis and angiogenesis via the overproduction of nitric oxide [42].

Our results demonstrate that CTSS knockdown inhibited COX-2 and iNOS expression, which subsequently prevented apoptosis and improved cell viability. TNF-α acts as a master regulator of inflammatory cytokine and ROS production; thus, it is necessary for both cell survival and death [43]. TNF-α protein levels in HG-treated samples were significantly higher than in the controls (Figure 2I); they were associated with increased ROS generation and cell apoptosis. Previous studies suggested ROS has a significant role in the synthesis of pro-inflammatory cytokines, such as TNF-α, IL-1β, etc. [44]. ROS causes the expression of pro-inflammatory cytokines by activating NF-κB pathways [45,46]. Besides the role of CTSS in inflammatory cytokine regulation, this study determined the effect of CTSS silencing on adhesion molecules such as VCAM-1 and ICAM-1. Similar to other studies, our study showed that hyperglycemia increased the expression of VCAM-1, ICAM-1, and VEGF-1 in HUVEC cells [29,47]. Hyperglycemia has been shown to increase the permeability of HUVEC monolayers, which is an early pathological mechanism in the progression of diabetic vascular complications [48]. MCP-1, VCAM-1, and ICAM-1 pro-inflammatory cytokines are expressed during ROS production and NF-κB translocation, which facilitate monocyte attachment to the vascular endothelium. Our findings showed that HG increased VEGFA expression in HUVECs while CTSS silencing significantly decreased VCAM-1, ICAM-1, and VEGFA expression (Figure 3A–D). VEGFA increases the adherence of leukocytes to vessel walls by upregulating the expression of ICAM-1 and VCAM-1. In addition, VEGFA is responsible for ED, angiogenesis, vascular permeability, and macular edema [49] by stimulating ECs through a protein kinase C (PKC)-dependent mechanism that subsequently enhances retinal and glomerular permeability [49,50]. In the current study, we found that HG levels increased VEGFA expression in HUVECs while CTSS silencing significantly decreased VCAM-1, ICAM-1, and VEGFA expression (Figure 3A–D). A close association has been observed between angiogenesis and inflammation under pathological conditions [51]. In addition to angiogenesis, VEGFA regulates HG-mediated increases in the expression of inflammatory cytokines IL-6 and MCP-1, as well as the adhesion molecule ICAM-1, via the NF-κB pathway activation. These factors contribute to angiogenesis and leukocyte recruitment [52]. In this study, significant tube formation was observed in HG-treated cells (Figure 3E–I), indicating that HG triggered angiogenesis and neovascularization in HUVECs. On the other hand, CTSS siRNA treatment under HG conditions suppressed tube formation as well as decreasing vascularization.

The complement (C) system is known to be an adaptive and innate immunity effector that is also responsible for neovascularization and angiogenesis [53,54]. Hyperglycemia induces C3 and C5 upregulation through NF-κB signaling, which releases pro-inflammatory cytokines and induces angiogenesis [55,56]. C3a and C5a activation can also cause macrophage-mediated angiogenesis via the release of VEGFA, IL-6, and TNF-α [53]. However, the relationship between CTSS and complement activation has not been fully elucidated. We demonstrated that CTSS knockdown suppressed the expression of C3 and C5 (Figure 4A,B) as well as the activation of C3a and C5a (Figure 4C,D) under hyperglycemic conditions. Overall, hyperglycemia-induced cytokine release and vascular complications in HUVEC cells are promising tools for the study of diabetic complications.

This study’s limitation include that we did not check whether the changes in CTSS caused by HG treatment were associated with hyperosmolarity or not. Further study of the relationship between high expressions of CTSS and hyperosmolarity would provide more robust data. Since the change in CTSS was already known for diabetic conditions, our study tried to reveal the immunomodulatory effect of CTSS silencing in an HG condition. More research is needed to understand the pathophysiological role of CTSS in hyperglycemia, but the findings of the current study provide strong evidence that CTSS plays a role in hyperglycemia-induced inflammation, angiogenesis, and vasculogenesis. CTSS silencing reduced the expression of pro-inflammatory cytokines, chemokines, and complement factors under hyperglycemic conditions (illustrated in Figure 5). These results suggest that CTSS may be considered a therapeutic target for the control and prevention of the development of hyperglycemic complications.

## 4. Materials and Methods

### 4.1. Antibodies and Chemical Reagents

Fetal bovine serum, Dulbecco’s phosphate-buffered saline, and 1X Trypsin-EDTA solution were obtained from Welgene (Gyeongsan-si, Gyeongsangbuk-do, Republic of Korea). Penicillin–streptomycin solution was acquired from HyClone Laboratories Inc. (South Logan, NY, USA). Dimethyl sulfoxide (DMSO) and 3-[4,5-dimethyl-2-thiazolyl]-2,5-diphenyltetrazoliumbromide (MTT) were obtained from Sigma-Aldrich (St. Louis, MO, USA). Paraformaldehyde (4%) was obtained from Biosesang (Seongnam-si, Gyeonggi-do, Republic of Korea), and Matrigel was purchased from BD Biosciences (Bedford, MA, USA). HUVECs (C2517A), endothelial basal medium-2 (EBMTM-2, CC-3156), and an endothelial cell growth medium-2 (EGMTM-2) BulletKit (CC-3162) were obtained from Lonza (Walkersville, MD, USA). Halt™ Protease and Phosphatase Inhibitor Cocktail, a Pierce™ RIPA Buffer (RIPA: radio-immunoprecipitation assay), and a Pierce™ BCA Protein Assay Kit were acquired from Thermo Fisher Scientific (Rockford, IL, USA). Opti-MEM™ reduced serum medium was also purchased from Thermo Fisher Scientific (Grand Island, NY, USA). Phenylmethylsulphonyl fluoride was purchased from Roche (Mannheim, Germany). Mounting medium with 4′,6-diamidino-2-phenylindole (DAPI)-Aqueous Fluoroshield and antibodies against VEGFA and COX-2 were acquired from Abcam (Cambridge, MA, USA). Antibodies against TNF-α were obtained from R&D Systems (Minneapolis, MN, USA). CTSS siRNA (h), antibodies against CTSS, ICAM-1, IL-6, and β-actin, and mouse anti-goat IgG-HRP, mouse anti-rabbit IgG-HRP, and goat anti-mouse IgG- conjugated with HRP were purchased from Santa Cruz Biotechnology (Dallas, TX, USA). Lipofectamine RNAiMAX reagent, calcein AM fluorescent dye, wheat germ agglutinin-Alexa Fluor™ 488 conjugate, Alexa Fluor 594 donkey anti-goat IgG (H&L), Alexa Fluor 594 goat anti-mouse IgG (H&L), and antibodies against MCP-1, NFκB p65, and iNOS were purchased from Invitrogen (Carlsbad, CA, USA). TB Green^®^ Premix Ex Taq™ was purchased from Takara Bio Inc. (Kusatsu, Shiga, Japan). The protein loading buffer and iScript cDNA synthesis kit were obtained from Bio-Rad Laboratories, Inc. (Hercules, CA, USA). The Ribospin II RNA extraction kit was purchased from GeneAll Biotechnology co., Ltd. (Songpa-gu, Seoul, Republic of Korea).

### 4.2. Cell Culture

HUVECs (Lonza, Walkersville, MD, USA) were cultured in EGM-2 growth kit-supplemented EBM-2 medium. A trypan blue exclusion assay was used to quantify mononuclear cells in small aliquots before seeding them into a 150 mm culture dish at a density of 1.0 × 10^6^ mL^−1^. This cell seeding density was used for subsequent experiments. HUVEC stocks were kept in a 37 °C incubator with 5% CO_2_.

### 4.3. Treatment of HUVECs with High Glucose Levels

HUVECs were seeded onto six-well tissue culture plates and incubated with EBM-2, supplemented with 2% FBS and 1% gentamicin, and maintained at 37 °C in a humidified incubator. For the HG treatments, final glucose concentrations of 15 mM, 30 mM, and 60 mM in EBM-2 were prepared and applied after 80% confluence was reached at 24 h and 48 h. The highest nontoxic dose and exposure time were selected for further experiments.

### 4.4. Small Interfering RNA (siRNA) Transfection

HUVECs were seeded with antibiotic-free culture medium containing EBM-2 (from the EGM-2 BulletKit) onto six-well tissue culture plates. After reaching 50–60% confluence, the cells were transfected with CTSS siRNA (h) (pool of 3 target-specific 19–25 nt siRNAs from Santa Cruz Biotechnology) and control siRNA at a final concentration of 10 nM using Lipofectamine RNAiMAX transfection reagent according to the manufacturer’s protocol. After transfection for 24 h, HUVECs were treated with 30 mM glucose for 24 h and then processed for further experiments.

### 4.5. Cell Viability and Proliferation Assay

In vitro cell viability under HG and after siRNA transfection was assessed via the MTT assay using a standard testing protocol [ISO10993-5:2009I]. The MTT assay was performed after 24 h and 48 h of treatment under different concentrations of HG as well as after 24 h of siRNA transfection. The cell viabilities (determined by measuring the optical density) were determined by following the previous protocol [57]. In brief, MTT solution (5 mg/mL in PBS) was added at a ratio of 1:9 to the cell culture media, followed by incubation for 4 h at 37 °C to allow for formazan crystal formation. Subsequently, dissolution was carried out in DMSO for 1 h to extract the formazan crystals. Finally, the absorbance was measured using an ELISA reader (EL, 312, Biokinetics reader; Bio-Tek Instruments, Winooski, VT, USA) at 595 nm. All samples were tested in triplicate.

In addition, according to the previous protocol [58], the growth and proliferation of siRNA-transfected HUVECs were observed using a fluorescence microscope (Leica DMi8, Wetzlar, Germany) after immunostaining the cells with wheat germ agglutinin-Alexa Fluor 488 conjugate for 10 min.

### 4.6. Quantitative RT-PCR

Total RNA was prepared after 24 h of HG treatment using a Ribospin II RNA extraction kit (GeneAll Biotechnology). Next, a NanoDrop-One spectrophotometer (Thermo Fisher Scientific, Cambridge, MA, USA) was used to measure the RNA concentrations, and the RNA was reverse-transcribed into cDNA using an iScript cDNA synthesis kit (Bio-Rad Laboratories, Inc. Hercules, CA, USA) in a SimpliAmp Thermal Cycler (Applied Biosystems, Singapore). All cDNA samples were kept at −20 °C for subsequent gene expression analysis. Using TB Green^®^ Premix Ex Taq™ (Takara Bio Inc., Kusatsu, Shiga, Japan) and a CFX96™ RT-PCR detection system, qRT-PCR was performed (Bio-Rad, Singapore). All real-time PCR experiments were performed in triplicate. The data were normalized against GAPDH levels, and the relative mRNA expression level was calculated using the 2^−△△Ct^ method. The nucleotide sequences of the primers used are provided in Table 1.

### 4.7. WB Analysis

Total protein was collected using a RIPA lysis buffer containing a 1X protease and phosphatase inhibitor cocktail. A BCA protein assay kit (Thermo Scientific, Rockford, IL, USA) was used to measure the total protein concentration. Protein samples in equal amounts were separated through 8% and 12% SDS-PAGE and transferred onto polyvinylidene difluoride membranes. Next, the membranes were incubated with specific primary antibodies overnight and subsequently incubated with specific secondary antibodies (HRP-conjugated). Chemiluminescent ECL reagents were used to detect the bands, and densitometric analyses were performed using a ChemiDoc™ XRS+ system (Bio-Rad, Hercules, CA, USA). As a loading control, β-actin was used.

### 4.8. Immunocytochemical Analysis

CTSS siRNA-transfected HUVECs were fixed in 4% paraformaldehyde solution after 24 h of HG treatment. The cells were permeabilized with 0.25% Triton X-100 and blocked with 2.5% bovine serum albumin (BSA). The cells were then immunostained overnight at 4 °C with antibodies against mouse CTSS (1:50), mouse C5a (1:50), and goat TNF-α (1:50). After incubation with the primary antibodies, the cells were incubated with the secondary antibody (Alexa Fluor 594, 1:1000). F-actin was stained with wheat germ agglutinin-Alexa Fluor 488 conjugate for 10 min. The cells were rinsed and mounted aqueously with Fluoroshield-DAPI. A fluorescence microscope (Leica DMi8) with Leica LAS AF software was used to visualize the images.

### 4.9. Tube Formation Assay

Pre-cooled 48-well plates were coated with Matrigel (phenol red free). The coating procedure was carried out on ice. siRNA-transfected and HG-treated HUVECs were harvested and diluted (1 × 10^7^ cells/mL) in EBM-2 medium containing low levels of serum. The treated HUVECs were then seeded on Matrigel-coated 48-well plates in triplicate and kept in a 37 °C incubator. Tube formation was observed from 4 h to 6 h. Images of tube formation were captured. To obtain fluorescent images of tube formation, 2 μM calcein AM solution was added, and the samples were then incubated at 37 °C for 20 min. Tube formation was observed, and a fluorescence microscope (Leica DMi8) was used to capture the phase-contrast and fluorescence images.

### 4.10. Statistical Analyses

All data are expressed as the mean ± SD (standard deviation), and all experiments were repeated at least three times. Significant differences were determined using Bonferroni’s multiple comparison test with a one-way analysis of variance (ANOVA) between the control and other groups. Statistically significant differences based on the multiple comparisons were defined and marked as follows: * *p* < 0.05, ** *p* < 0.01, *** *p* < 0.001 versus control samples; # *p* < 0.05, ## *p* < 0.01, ### *p* < 0.001 versus HG-treated samples. GraphPad Prism 8 was used to perform all statistical analyses.

## Figures and Tables

**Figure 1 ijms-24-05428-f001:**
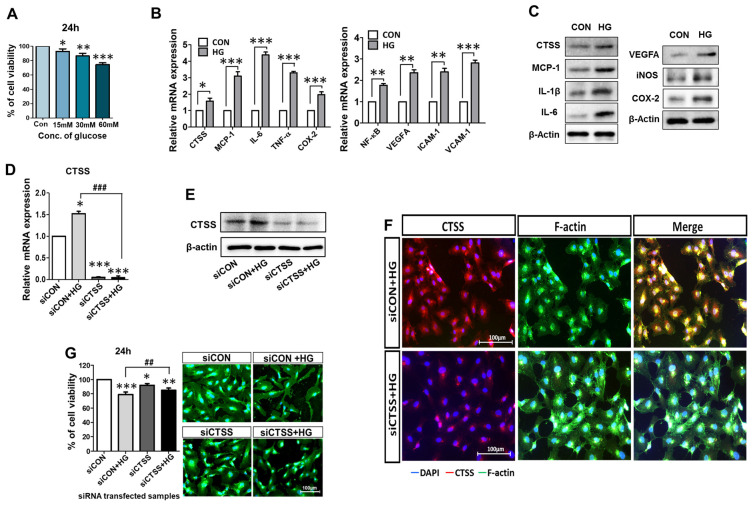
Upregulation of pro-inflammatory signals by glucose treatment (30 mM) and evaluation of CTSS knockdown in HUVECs. (**A**) HUVECs viability was assessed after treatment with different concentrations of glucose (15, 30, and 60 mM) for 24 h. (**B**,**C**) High expression levels of mRNA and protein of CTSS, pro-inflammatory factors, and vascular endothelial markers in samples treated with high glucose (HG) compared to control samples. (**D**,**E**) mRNA expression and Western blot analysis of CTSS expression after siRNA transfection. (**F**) Immunofluorescence analysis of CTSS expression. (**G**) HUVEC viability after siRNA transfection under hyperglycemic conditions was assessed by MTT assay and fluorescence staining. Values are expressed as mean ± SEM (*n* = 3 biological replicates per group). *p* < 0.05 indicates statistical significance. * *p* < 0.05, ** *p* < 0.01, *** *p* < 0.001 versus control samples; ## *p* < 0.01, ### *p* < 0.001 versus HG-treated samples.

**Figure 2 ijms-24-05428-f002:**
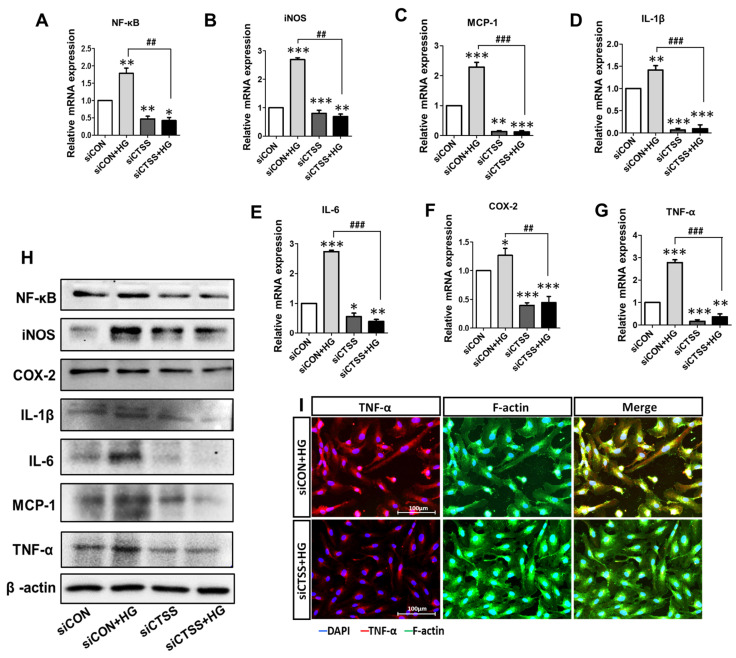
High expression of pro-inflammatory genes and proteins after glucose treatment and low expression observed after cathepsin S (CTSS) knockdown. (**A**–**G**) mRNA levels of pro-inflammatory gene markers. (**H**) Protein expression of pro-inflammatory chemokines and cytokines. (**I**) Immunofluorescence analysis of TNF-α protein expression. Values are expressed as mean ± SEM (*n* = 3 biological replicates per group). *p* < 0.05 indicates statistical significance. * *p* < 0.05, ** *p* < 0.01, *** *p* < 0.001 versus control samples; ## *p* < 0.01, ### *p* < 0.001 versus HG-treated samples.

**Figure 3 ijms-24-05428-f003:**
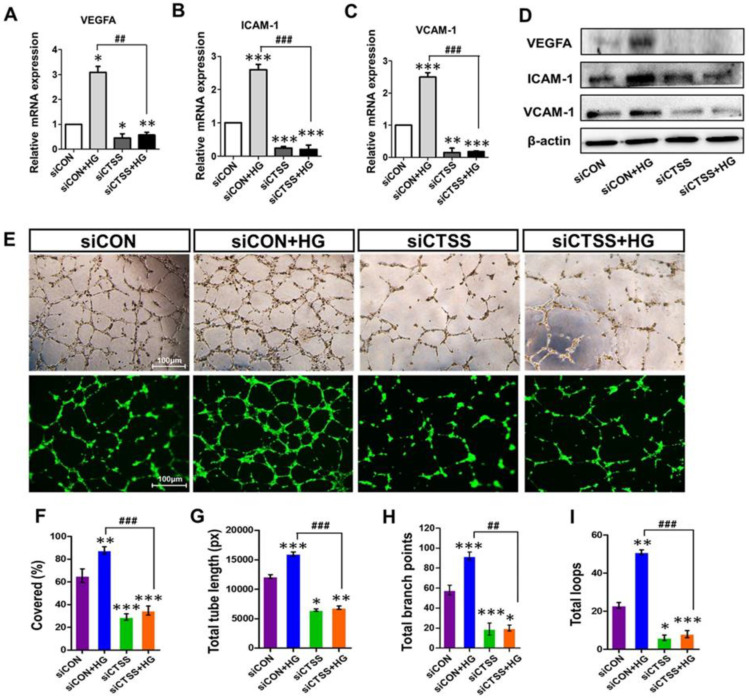
Expression of vascular endothelial growth factor A (VEGFA) and adhesion molecules ICAM-1 and VCAM-1, as well as the tube formation ability of cathepsin S (CTSS) siRNA-transfected HUVEC cells post-treatment with 30 mM high glucose. (**A**–**D**) mRNA and protein expression of VEGFA, ICAM-1, and VCAM-1. (**E**) Phase-contrast images of tube formation in HUVECs and fluorescence images of tube formation captured after calcein AM staining. Increased tube formation was observed in siCON + HG samples compared to siCTSS + HG samples. (**F**) Percentage covered area (%), (**G**) total tube length (measured in pixels, px), (**H**) number of branching points, and (**I**) number of total loops of tube formation were obtained by ImageJ densitometry. Values are expressed as mean ± SEM (*n* = 3 biological replicates per group). *p* ≤ 0.05 indicates statistical significance. * *p* < 0.05, ** *p* < 0.01, *** *p* < 0.001 versus control samples; ## *p* < 0.01, ### *p* < 0.001 versus HG-treated samples.

**Figure 4 ijms-24-05428-f004:**
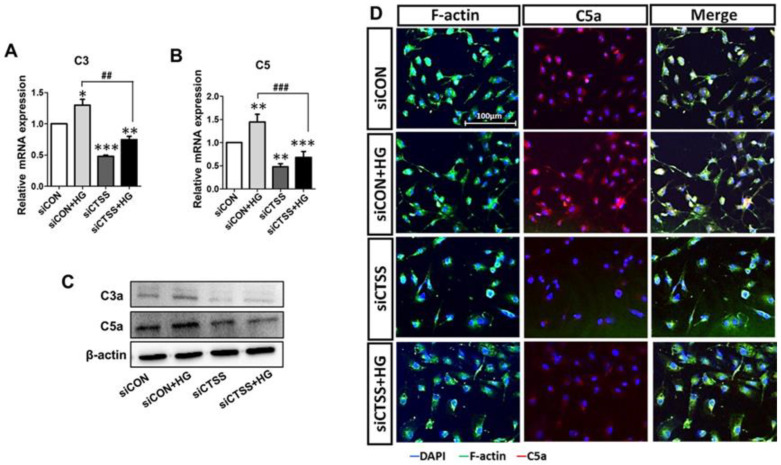
Treatment with cathepsin S (CTSS) siRNA suppressed complement factor activation. (**A**,**B**) Expression of complement C3 and C5 genes and (**C**) C3a and C5a proteins after CTSS siRNA transfection under high glucose conditions. (**D**) Immunofluorescence analysis of C5a. Values are expressed as mean ± SEM (*n* = 3 biological replicates per group). *p* < 0.05 indicates statistical significance. * *p* < 0.05, ** *p* < 0.01, *** *p* < 0.001 versus control samples; ## *p* < 0.01, ### *p* < 0.001 versus HG-treated samples.

**Figure 5 ijms-24-05428-f005:**
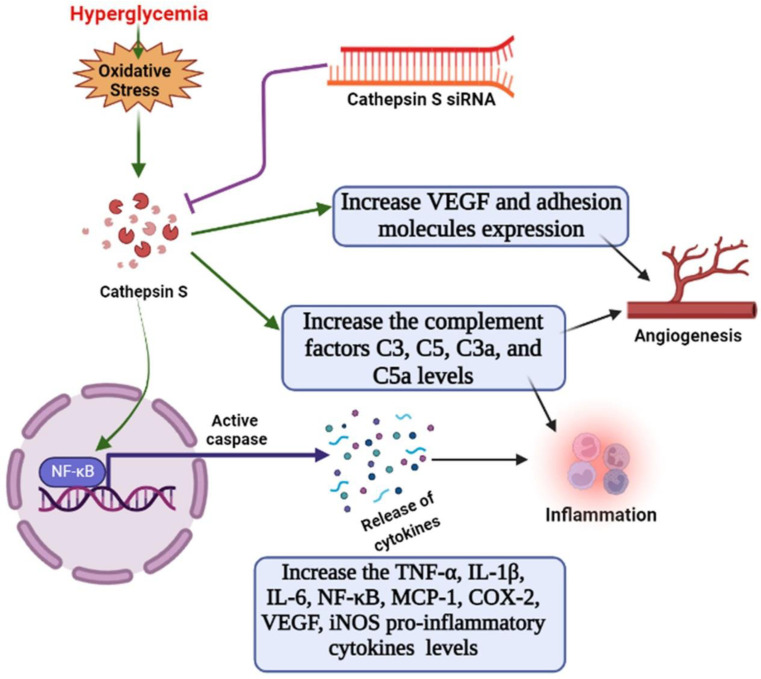
Schematic diagram. The role of cathepsin S in the activation of angiogenesis, complement factors, and endothelial inflammation via the NF-κB pathway under hyperglycemic conditions in HUVECs, as well as the suppressive effects of cathepsin-siRNA on inflammatory responses.

**Table 1 ijms-24-05428-t001:** qRT-PCR primer sequences used in this study.

Gene	Sequence 5′-3′
CTSS	F-GCCTGATTCTGTGGACTGG
	R-GATGTACTGGAAAGCCGTTGT
VEGFA	F-GGTGCCCGCTGCTGTCTAAT
	R-TGCAACGCGAGTCTGTGTTT
TNF-α	F-AAGCTGAGGGGCAGCTCCAGT
	R-TCTGGTAGGAGACGGCGATGC
ICAM-1	F-CTTCGTGTCCTGTATGGCCC
	R-CACATTGGAGTCTGCTGGGA
VCAM-1	F-GTCAATGTTGCCCCCAGAGATA
	R-ACAGGATTTTCGGAGCAGGA
iNOS	F-GCTCTACACCTCCAATGTGACC
	R-CTGCCGAGATTTGAGCCTCATG
NFκB	F-CACTTATGGACAACTATGAGGTCTCTGG
	R-CTGTCTTGTGGACAACGCAGTGGAATTTTAGG
Cox-2	F-TTGCTGGCAGGGTTGCTGGTGGTA
	R-CATCTGCCTGCTCTGGTCAATGGAA
IL-1β	F-TTGCTCAAGTGTCTGAAGCAGC
	R-CTTGCTGTAGTGGTGGTCGG
IL-6	F-GGATTCAATGAGGAGACTTGCC
	R-GGGTCAGGGGTGGTTATTGC
MCP-1	F-GGCTGAGACTAACCCAGAAAC
	R-GAATGAAGGTGGCTGCTATGA
C3	F-ACCAGCAGACCGTAACCATC
	R-GCAGCCTTGACTTCCACTTC
C5	F-CTCCTCAGGCCATGTTCATT
	R-CTCCAGGCAATTGTTTTGGT
GAPDH	F-GAGAAACCTGCCAAGTATGATGAC
	R-AGAGTGGGAGTTGCTGTTGAAG

F forward, R reverse.

## Data Availability

The data that support the findings of this study are available from the corresponding author upon reasonable request.

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
