# Peer review of "Cathepsin S Knockdown Suppresses Endothelial Inflammation, Angiogenesis, and Complement Protein Activity under Hyperglycemic Conditions In Vitro by Inhibiting NF-κB Signaling"

_ijms, 2023, doi:10.3390/ijms24065428_

Round 1

Reviewer 1 Report

The title of the manuscript, Cathepsin S knockdown suppresses endothelial-inflammations, angiogenesis and complements under hyperglycemia via inhibiting NF-κB signaling in vitro’ is appropriate for this study. The findings of the study are also interesting and novel. However, there are several concerns as follows that need to be addressed:

1.      Including a brief hypothesis of the current study will make the abstract more meaningful.

2.      In all figures, the authors need to mention the exact number of biological replicates used for each type of experiment (n=?).

3.      In figure 2, why did the author provide the immunofluorescence staining only for the TNF-α protein expression? They must provide the immunofluorescence staining data for all the other proteins, including NF-κB, iNOS, COX-2, IL-1β, IL-6 & MCP-1.

4.      Type 1 diabetes mellitus (DM) is associated with elevated blood glucose. However, type-2 DM is associated with elevated blood glucose and insulin levels. In this study, the authors did not provide any justification for the significance of their study that supports a specific type of DM. If the authors want to emphasize type-1 DM then they need to provide a sufficient literature review that supports their hypothesis/findings associated with type-1 DM. On the other hand, if they want to emphasize type-2 DM, then they need to include high insulin levels with high glucose to mimic type-2 DM condition.

5.      Hyperglycemia-induced oxidative stress is associated with an increased formation of reactive species, including the reactive oxygen species, reactive carbonyl species, and reactive nitrogen species. Therefore, measuring the levels of some reactive molecules, including ROS, malondialdehyde, 4-hydroxy-2-nonenal protein adducts will make the findings of this study more concrete.

6.      There are several grammatical mistakes throughout the whole manuscript.

Reviewer 2 Report

The paper from Sayed et al., present the impact of addition of 30 mM glucose (Hyperglicemic condition) on HUVEC cells and showed that it induce the upregulation of many proteins linked to inflammation and angiogenesis. This effect is totally attenuated by limiting cathepsin S expression. In this paper hyperglycemic conditon decrease cell viability, increase tube formation and angiogenesis by HUVEC and increase numerous pro-inflammatory cytokines. The study is quite descriptive and do not propose any causal or mechanism to integrate all the modifications induces by hyperglycemia with or with out Cathepsin S siRNA.

The main concern rely on the impact of 30mM glucose on osmolarity. Since cathepsin S has been shown to be stimulated by hyperosmolarity. The author should provide a control for such situation showing that the increase in osmolarity is not main factor in what they see. Addition of a sugar or polyol like sorbitol with the same osmolarity and not involved in hyperglycemia? does it increase in CTSS and downstream targets?

What is the mechanism leading to cell toxicity while at the same time tube formation is improved by 30mM glucose. Is it real cytotoxicity and not a bias of MTT assay in a context of hyperglycemia or hyprosmolarity affecting mitochondrial function and metabolism?

The impact of CTSS knock-down is far more effective by itself compared to control than the impact of hyperglycemia. The use of a pharmacological inhibitor of Cathepsin S with the right concentration to block the increased activity could be more relevant to specifically address the modification induced by 30mM glucose.

Title of Fig1 is too long and not clearly understandable.

Fig1G upper right panel shoulb be siCon+HG?

Fig5 extrapolate a lot from the data in the paper. No data on ROS production or damaged lysosome were provided. Unless the authors provide somme data in these area, the figure should be adapted to stick more to the data in the paper.

Round 2

Reviewer 1 Report

Thank's to the authors for addressing all of my concerns I had with the preliminary version of the manuscript. I recommend the current version of the manuscript should be published. 

Author Response

Thank you so much for recommending our revised manuscript for publication. We would like to express our gratitude to the reviewer for taking time to review our manuscript. Your comments and suggestions were greatly valued in revising the manuscript.

Reviewer 2 Report

Dear authors,
Thank you for the changes to the article. I still cannot believe in the absence of impact of osmolarity until there is a control suitable for this conclusion.
The Jain et al. reference you cite uses mannitol but does not present any data on mannitol (data not shown). The condition of hyperosmolarity as referenced in this article (Hyperosmolar hyperglycemic syndrome.Adeyinka A, Kondamudi NP.2022; PMID: 29489232) states that:
According to the recommendation of the American Diabetic Association and
 current international guidelines, HHS is defined by a plasma glucose level greater than 600 mg/dL, an effective plasma osmolarity greater than 320 mOsm/L, and the absence of significant ketoacidosis.
600 mg/dL corresponds to about 33mM glucose and hyperosmolarity is not a binary process (no biological impact of increasing osmolarity below 33mM and suddenly hyperosmolarity above!).
So a control with 30mM mannitol is needed at least for a primary outcome like the level of cathepsin expression in your experimental conditions. This will help rule out (or not) possible regulation involving the already described role of hyperosmolarity.
Without such a control, your data do not fully support your conclusions.
So either provide suitable controls or modify the title abstract and discussion discussion to specify that you cannot exclude the effect of
hyperosmolarity in the observed results.

Round 3

Reviewer 2 Report

no more comments